# Design, Optimization, and Correlation of In Vitro/In Vivo Disintegration of Novel Fast Orally Disintegrating Tablet of High Dose Metformin Hydrochloride Using Moisture Activated Dry Granulation Process and Quality by Design Approach

**DOI:** 10.3390/pharmaceutics12070598

**Published:** 2020-06-27

**Authors:** Alhussain H. Aodah, Mohamed H. Fayed, Ahmed Alalaiwe, Bader B. Alsulays, Mohammed F. Aldawsari, El-Sayed Khafagy

**Affiliations:** 1Department of Pharmaceutics, College of Pharmacy, Prince Sattam Bin Abdulaziz University, Al-kharj 11942, Saudi Arabia; m.fayed@psau.edu.sa (M.H.F.); a.alalaiwe@psau.edu.sa (A.A.); b.alsulays@psau.edu.sa (B.B.A.); moh.aldawsari@psau.edu.sa (M.F.A.); e.khafagy@psau.edu.sa (E.-S.K.); 2Kayyali Chair for Pharmaceutical Industries, Department of Pharmaceutics, College of Pharmacy, King Saud University, Riyadh 11451, Saudi Arabia; 3Department of Pharmaceutics and Industrial Pharmacy, Faculty of Pharmacy, Suez Canal University, Ismailia 41522, Egypt

**Keywords:** metformin hydrochloride, orally disintegrating tablet (ODT), quality-by-design (QbD), moisture activated dry granulation (MADG), high drug loading

## Abstract

Compression of cohesive, poorly compactable, and high-dose metformin hydrochloride into the orally disintegrating tablet (ODT) is challenging. The objective of this study was to develop metformin ODT using the moisture activated dry granulation (MADG) process. There are no reports in the literature regarding the development of ODT based on MADG technology. The feasibility of developing metformin ODT was assessed utilizing a 3^2^ full factorial design to elucidate the influence of water amount (X_1_) and the amount of pregelatinized starch (PGS; X_2_) as independent variables on key granules and tablets’ characteristics. The prepared granules and tablets were characterized for granule size, bulk density, flow properties, tablets’ weight variation, breaking force, friability, capping tendency, in vitro and in vivo disintegration, and drug release. Regression analysis showed that X_1_ and X_2_ had a significant (*p* ≤ 0.05) impact on key granules and tablets’ properties with a predominant effect of the water amount. Otherwise, the amount of PGS had a pronounced effect on tablet disintegration. Optimized ODT was found to show better mechanical strength, low friability, and short disintegration time in the oral cavity. Finally, this technique is expected to provide better ODT for many kinds of high-dose drugs that can improve the quality of life of patients.

## 1. Introduction

Metformin HCL (metformin), “chemically designated as 1,1-dimethylbiguanide hydrochloride”, is an oral hypoglycemic agent widely used in the treatment of type-2 diabetes mellitus (T2DM), specifically in overweight and obese individuals [1,2]. The hypoglycemic effect of metformin is owing to its ability in the suppression of hepatic glucose production and increases the peripheral sensitivity to insulin with higher safety [3]. Thanks to its low price, safety, potential cardiovascular protection, and no incidence of weight gain, metformin has been approved for initial therapy of T2DM [4,5]. Recently, clinical studies have indicated that metformin may improve outcomes of several cancers and reduce cancer risk in diabetic patients [6]. Some other studies showed that metformin might extend the lifespan of people, which led to the ongoing clinical trial “Targeting Aging with Metformin”, aimed at lateness, the process of aging and extend healthy lifespan [7,8].

Metformin is a BCS class-III drug, with high water solubility and low permeability to cell membrane [9]. The usual dose for immediate-release tablet of metformin is 250–500 mg three times daily up to 3 g per day [10]. The high dose of metformin needs a large tablet size, which decreases compliance in geriatric patients owing to the difficulty in swallowing [10,11]. To overcome swallowing problems, the conventional tablets of metformin are chewed or splitted [12]. However, metformin has a strong bitter taste, and these procedures increase bitter taste recognition, which leads to patient rejection [13]. Consequently, metformin orally disintegrating tablet (ODT) is a preferred dosage form for diabetic patients with swallowing issues, because the fast disintegration in the patient’s oral cavity allows easy swallowing without additional water [10,13].

However, the development of metformin ODT using the most economical direct compression method encounters a challenge owing to its poor compressibility, moisture sensitivity, and great tendency for capping [14]. To overcome this challenge, the formulator has to use the wet granulation process to get agglomerates of drug and excipients with adequate compression characteristics [15]. This may lead to formation of solid bridges ”necks” among metformin particles that impart mechanical strength, lack powder flow, and sticking issues through the tableting process owing to consequent moisture migration or desorption during the wet granulation process [16,17]. In addition, from an industrial perspective, the wet granulation method is inferior to a direct compression and dry granulation methods because an additional drying step in a separate equipment like fluidized-bed dryer and tray dryer is necessary after the agglomeration phase [18,19]. Furthermore, adhesion of produced granules on granulator walls as well as granule collapse by mechanical stress demonstrate process challenges during the drying stage [19]. To overcome the aforementioned issues, a novel continuous granulation technique (one-pot process) called moisture activated dry granulation (MADG) was characterized by Ullah et al. [20,21].

MADG is a straightforward, innovative technique and the whole process could be achieved within a usual high-shear mixer/granulator; thus, this technology is described as ‘‘one-pot granulation process” [21]. The MADG process can be classified into two distinctive steps: (1) the agglomeration step and (2) the moisture absorption step. Initially, drug and functional excipients like binders and disintegrant are pre-blended in conventional high shear mixer/granulator and the binder is activated using a small amount of water (1–4% m/m) to form the agglomerates. During the absorption stage, the moisture of obtained granules is absorbed by adding an absorbent powder, resulting in the formation of dry and free flowing granules [22].

Compared with popular wet granulation methods like high shear granulation (HSG) and fluidized bed granulation (FBG), MADG is considered a promising alternative technique to combine the benefits of HSG, while avoiding the issues associated with the drying step described above [18,22]. Besides, HSG and FBG require the discharge and filling of intermediate granules for cascading the manufacturing process. However, these transfer steps can be avoided in the MADG process, which saves processing time and reduces the risk of exposure to potentially highly potent compounds [19]. Furthermore, MADG has very few variables, resulting in no issue of endpoint sensitivity and lesser need for costly Process Analytical Techniques (PAT) technology [21]. In many cases, HSG of drug substances are difficult, particularly when compounds with poor solubility, wettability, and small particle size at drug loading are above 70%. Therefore, MADG is an ideal granulation process for the granulation of hydrophobic materials [21].

There are no reports in the literature regarding the development of ODT using the MADG technique. Therefore, the primary objective of this research was to investigate the feasibility of producing ODT containing a high dose (500 mg) of metformin, based on the MADG process. The secondary objective of the present investigation was to demonstrate the application of quality-by-design (QbD) approach for the development and optimization of metformin ODT in terms of acceptable mechanical strength and rapid disintegration in oral cavity. Therefore, this dosage form is good to improve compliance of diabetic patients as well as avoiding difficulties in swallowing.

Quality by design (QbD) is an efficient, systematic and knowledge-based approach, which is applied to improve the quality attributes of pharmaceutical products. The design of experiment (DoE) is a key element of QbD. It is applied to identify the main variables affecting the critical quality attributes in early product development (screening step) and to determine the values of the variables maximizing product and process performance (optimization step) [23]. The elements of the QbD framework are described in International Council for Harmonization (ICH) Q8. QbD elements include the following process: (1) the definition of a quality target product profile (QTPP); (2) the identification of the critical quality attributes (CQAs) of the drug product that are directly related to the QTPP; (3) the identification of critical material attributes (CMAs) and (4) the identification of critical process parameters (CPPs) linked to CQA; (5) the definition of a design space (DS); and (6) the identification of a control strategy that includes specifications for the drug substance(s), excipient(s), and drug product as well as controls for each step of the manufacturing process that allows a continual improvement [23,24]. The QTPP is defined as a “prospective summary of the quality characteristics of pharmaceutical product that will be achieved to ensure the desired quality including efficacy and safety of drug product” [25]. A CQA can be defined as “a physical, chemical, biological, or microbiological property or characteristic that should be within an appropriate limit, range, or distribution to ensure the desired product quality” [25]. QTPP and CQAs for metformin fast orally disintegrating tablets are listed in Table 1.

## 2. Materials and Methods

### 2.1. Materials

Metformin hydrochloride (CNLAB Canada, Asian Group, Co., Shaanxi, China); D-mannitol, Mannogem^®^ (SPI Pharma, Wilmington, NC, USA); partially pregelatinized starch, Starch 1500^®^ (Colorcon, Dartford, UK); colloidal silicon dioxide, Aerosil 200^®^ (Evonic, Hanau-Wolfgang, Germany); and sodium stearyl fumarate, PRUV ^®^ (JRS pharma, Rosenberg, Germany).

### 2.2. Experimental Design

Factorial designs (fractional and full factorial) are applied to examine two or more factors in a test. Full factorial is the most frequently employed screening design. In a full factorial design, all possible combinations of factors at all levels are investigated. This approach is a very cost-effective way of obtaining the maximum amount of information with the minimum experimental effort [23].

Before application of the design, the number of preliminary trials was done to define the independent variables as well as the variables ranges at which granules and tablets with reasonable quality were produced. A two-factor three-level (3^2^) full factorial design was applied using Design-Expert software (Version-12, State-ease, Inc., Minneapolis, MN, USA) to examine the influence of the water amount (X_1_; 1–4% *w/w*) and the pregelatinized starch (PGS) amount (X_2_; 5–15% *w/w*) as independent variables on the key characteristics of granules and corresponding ODT. As shown in Table 2, each independent variable was examined at three levels, expressed as low (−1), medium (0), and high (+1). Table 3 shows the entire design matrix including nine experiments. All experiments were run and investigated in triplicate. To confirm the validity of the design, the run at the center point was repeated five times on several days, and the average values of these experiments show good reproducibility of the process. All obtained results were expressed as a mean ± SD. Statistical analysis of obtained results was performed by the analysis of variance (ANOVA) test using Design-Expert software.

### 2.3. Selection of Excipients

Pharmaceutical excipients have a significant influence on tablet properties. Thus, selection of excipients and their amount is a critical need, and depends on the drug properties, target dosage form, and manufacturing process [26]. Mannitol is a preferred excipient for developing ODT of the moisture-sensitive drugs like metformin owing to its non-hygroscopic nature, compactibility, sweet taste, and cool feeling that it leaves in the mouth [27]. Besides, it reduces disintegration time in the oral cavity owing to its higher solubility compared with other water-soluble excipients used in the preparation of ODT [28]. PGS was chosen in the present study because it shows dual functionality as binder/disintegrant. It could exhibit binding properties in the granulation applications, and then it would become a strong disintegrant when exposed to water owing to its ability to induce swelling. In addition, PGS had lower propensity for moisture uptake than sodium starch glycolate and croscarmellose sodium and drew less moisture into tablets [29]. Thus, it is a suitable binder/disintegrant for moisture sensitive drugs in granulation applications. Regarding colloidal silicon dioxide, most studies claim that it is a highly suitable absorbent for MADG as it has a very low water content [18]. Finally, sodium stearyl fumarate was preferable as a lubricant in the preparation of ODT because it has a great lubrication effect without obviously reducing other critical tablet attributes, including tablet mechanical strength, disintegration time, and drug release [30].

### 2.4. Preparation of Granules and Tablets

Table 4 shows the formulations used in the present investigation. Granulation experiments were carried out in conventional high-shear granulator (BOSCH Packaging Technology, Schopfheim, Germany). The powder formulation (metformin, mannitol, and binder) was dry mixed in the granulator for 2 min at a high impeller and chopper speed. The powder blend was then granulated by addition of the specified amount of water using a binary spray nozzle. After addition of the water amount, the blend was wet massed for a specified massing time of 2.5 min. For the absorption stage (2 min), the moisture absorbent colloidal silicon dioxide was added when the chopper was stopped. Finally, pre-sieved lubricant sodium stearyl fumarate was blended directly in the granulator for 1 min at the low impeller speed of 250 rpm. The lubricated blend was then compressed at a compression force of 13 KN into 625 mg tablets using a fully automated rotary tablet press (RoTap-T 2.0, Kg pharma, Berlin, Germany). The tablet press was set up to produce eight tablets per compression cycle using standard 10 mm flat tooling. The prepared tablets were collected and stored for subsequent evaluation.

### 2.5. Granules Characterization

#### 2.5.1. Mean Granule Size

The mean granule size was measured by the laser diffraction method using (Mastersizer 2000, Malvern Instruments Ltd., Malvern, Worcestershire, UK).

#### 2.5.2. Granules Bulk Density

Bulk density was measured according to the method stated in United States Pharmacopeia (USP 42-NF37). The granules sample (n = 3) were gently poured into a 25 cm^3^ graduated cylinder up to particular volume (V_b_). The mass of granules (m) was then determined and the bulk density (ρ_b_) was calculated using Equation (1).
(1)$$\rho_{b}=\frac{m}{V_{b}}$$

#### 2.5.3. Granules’ Flow

The flow properties of the prepared granules were performed using the angle of repose method, as stated in USP 42-NF37. The granules sample (n = 3) was carefully poured through a dry funnel located at 2 cm (H) above a circular plate to form a conical heap. The diameter (D) of the granules heap was determined, and angle of repose was calculated using Equation (2).
(2)$$\tan\;(\theta )=\frac{2H}{D}$$

### 2.6. Tablets’ Characterization

#### 2.6.1. Tablets’ Weight Variation

This test was done following the procedure outlined in USP 42-NF37. The weight of twenty random tablets (n = 20) was individually measured using digital analytical balance (Shimadzu, UP series, Kyoto, Japan), and the average tablet weight and corresponding standard deviation (SD) were determined.

#### 2.6.2. Tablets’ Breaking Force

Breaking force is a measurement applied to indicate the tablet strength. The breaking force for ten random tablets (n = 10) was individually measured using (PharmaTest, PTP 111 EP, Hainburg, Germany). The mean and corresponding SD of tablets breaking force were determined.

#### 2.6.3. Tablets’ Friability and Percent Capping

Friability test is another measurement used to indicate the strength of prepared tablets. This test was done according to the procedure described in USP 42-NF37. Ten tablets (n = 10) were randomly chosen, accurately weighed (*W_1_*), placed in an automatic tablet friabilitor (PharmaTest, PTF 300, Hainburg, Germany), and rotated at 25 rpm for 4 min. The tablets were removed, de-dusted, and accurately weighted (*W_2_*). Friability as the percentage of mass loss was calculated using Equation (3).
(3)$$Friability=\frac{W_{1}-W_{2}}{W_{1}}\times 100$$

Upon dusting the tablets following friability testing, the number of tablets that showed capping was determined. Capping of the tablets was reported as the percent of tablets capped out of the total tested tablets [31].

#### 2.6.4. In Vitro Tablets’ Disintegration

In vitro disintegration test was performed according to the USP 42-NF37 procedure, in a compendial disintegration tester (PharmaTest, PTZ-S series, Hainburg, Germany), using 800 mL distilled water at 37 ± 0.5 °C as disintegration medium. Disintegration time (DT) of randomly selected six tablets (n = 6) was recorded in seconds. The average DT and corresponding SD were determined.

#### 2.6.5. Tablets’ Disintegration in the Oral Cavity

Tablet disintegration in the oral cavity was investigated in six healthy subjects after providing written consent. Prior to the test, the mouth of each subject was rinsed three times with 150 mL of the pure water. The tablet was placed on the tongue of each subject without crushing until the tablet had disintegrated completely. The time required for complete tablet disintegration was recorded in seconds [32]. This study was approved by the Ethics Committee of College of Medicine, Prince Sattam Bin Abdulaziz University.

#### 2.6.6. In Vitro Drug Release

The in vitro drug release was carried out using the USP paddle method utilizing dissolution apparatus (Distek 2500, Distek Inc., North Brunswick Township, NJ, USA) under the following conditions: 900 mL distilled water adjusted at 37 ± 0.5 °C was used as a release medium and the paddles’ speed was set to 100 rpm. The samples were analyzed in the release medium at specified time interval of 5 min up to 30 min using in situ fiber optic UV testing (Distek Opt-Dis 410, Distek Inc., North Brunswick, NJ, USA) at λ_max_ of 232 nm. A calibration curve was generated using pure metformin solution at a concentration range of 0.025–0.15 mg·mL^−1^ [12]. The present compendial dissolution procedure was not initially designed to be biorelevant for the oral cavity, but for comparative study.

## 3. Results and Discussion

### 3.1. Data Fitting to the Model

In the case of the 3^2^ full factorial design, the following quadratic equation (Equation (4)) was fitted to the obtained results to demonstrate the effect of independent variables on the measured responses of prepared granules and corresponding ODT.
*Y* = *β*_0_ + *β*_1_ X_1_ + *β*_2_ X_2_ + *β*_3_ X_1_X_2_ + *β*_4_ X_1_^2^ + *β*_5_ X_2_^2^(4)
where *Y* is the dependent variable; *β*_0_ is the arithmetic mean of nine experiments, *β*_1_ to *β*_5_ are the coefficient estimates of independent variables. X_1_ and X_2_ are the individual effects, X_1_X_2_ is the interaction effect, and X_1_^2^ and X_2_^2^ are the quadratic effects. To evaluate the validity of the selected design, the experimental values were quantitatively compared with the predicted values of the dependent responses and the relative error (%) was measured using the following equation (Equation (5)) [33].
(5)$$Relative\;error\;\left(\%\right)=\left(\;\frac{\left|Predicted\;value-Experiment\;value\right|}{Predicted\;value}\;\right)\times 100$$

As shown in Table 5, the regression models performed reasonably well, with suitable correlation coefficients (R^2^) of higher than 0.9343. Additionally, good agreement was observed between adjusted R^2^ and predicted R^2^, confirming that the obtained results were well fitted by the regression models. The generated mathematical models were helpful for recognizing the impact of the independent variables on the dependent response through quantitative comparison of the variable coefficients [34].

### 3.2. Effect of Independent Variables on Granules’ Characteristics

#### 3.2.1. Mean Granules’ Size

Granulations with different amounts of water and PGS were found to be desirable as no bowl wall adhesion or formation of big lumps occurred. Granule properties are summarized in Table 6. It was observed that increasing the water amount from 1% to 4% *w/w* and the PGS from 5% to 15% *w/w* resulted in an increase of mean granule size (D_50_) from 221.15 to 520.39 μm, as well as a decline in the percent fines from 26.42% to 2.36%. On the basis of the results of granules’ characteristics, regression models of granule size and percent fines were generated. The results of regression analysis are listed in Table 7.

The ANOVA test showed that the D_50_ was significantly influenced by the water amount (*p* < 0.0001) and the amount of PGS (*p* = 0.0033), with predominant effect of the water amount, as shown by its high sum of squares compared with the PGS amount (62,230.35 for the water amount and 11,150.83 for the PGS amount). Specifically, the D_50_ was found to be positively correlated with the water amount and the amount of PGS with respect to the sign of their coefficient estimates (+101.84 for the water amount and +43.11 for the PGS amount). As shown in Figure 1A, the D_50_ was increased in a response to an increase in the water amount and the amount of PGS. On the other hand, both the water amount and the amount of PGS had a significant (*p* < 0.0001 for the water amount and *p* < 0.0001 for the PGS amount) effect on the percent fines, with the dominant effect of the water amount, as shown by its higher sum of squares compared with the PGS amount (499.05 for the water amount and 55.09 for the PGS amount). Furthermore, coefficient estimates values revealed that change in the water amount had higher impact on percent fines in a negative direction (coefficient estimate of water amount = −9.12), while the PGS amount with a low coefficient estimate had a small effect in the same direction (coefficient estimate of the PGS amount = −3.03). This suggests that granulation with high amounts of water and PGS results in producing a low amount of fines. Increasing the water amount promotes proper wetting of the particles surfaces that promote the binding between particles and growth through formation of liquid bridges among the particles [35]. Moreover, at a high amount of added water, more liquid was available to promote starch gelatinization. This gelatinization activated the binding properties of starch, leading to an increase in the granule size [36]. Takasaki et al. reported that the main mechanism for granulation using MADG is binding between particles in the presence of mechanical shear [37].

The analysis of variance demonstrated that the linear model is valid with a significant *p*-value (*p* < 0.0001 for mean granule size and *p* < 0.0001 for percent fines) and insignificant lack of fit value (F = 64.06 for mean granule size and F = 577.35 for percent fines). The linear model equations that explained the effect of amounts of water and PGS on mean granule size and percent fines can be expressed as follows:

Mean granule size (µm) = 373.16 + 101.84 × X_1_ + 43.11 × X_2_

Percent fines (%) = 14.42 − 6.12 × X_1_ − 3.03 × X_2_

#### 3.2.2. Granules Bulk Density

The influence of the amounts of water and PGS on granules density is shown in Table 6. It was observed that the bulk density of the prepared granules increased from 0.288 to 0.412 g·cm^−3^ as the water amount and the PGS amount increased. As shown in Table 7, the ANOVA showed that the amounts of water and PGS had a significant (*p* < 0.0001 for the water amount and *p* = 0.0006 for the PGS amount) positive effect on granules’ bulk density with respect to the sign of their coefficient estimates (+0.0363 for the water amount and +0.0250 for the PGS amount). However, the water amount was the most significant variable according to the magnitudes of its sum of squares (0.0079 for the water amount and 0.0038 for the PGS amount). Otherwise, the mutual interaction between the water amount and the amount of PGS also had a significant (*p* = 0.0395) effect on granules’ bulk density. The 3D response surface plot shown in Figure 1C demonstrates the relationship between the variables influencing granules’ bulk density. It displays positive effects for the water amount and the amount of PGS on the bulk density of the produced granules.

The analysis of variance demonstrated that the 2FI model is valid with a significant *p*-value (*p* = 0.0002) and insignificant lack of fit value (F = 64.06). The 2FI model equation that explained the effect of water amount and PGS amount on granules bulk density can expressed as follows:

Bulk density (g·cm^−3^) = 0.3399 + 0.0363 × X_1_ + 0.0250 × X_2_ + 0.0110 × X_1_X_2_

#### 3.2.3. Granules’ Flow

As shown in Table 6, the angle of repose of prepared granules decreased from 33.45° to 25.83° as the water amount and the PGS amount increased, suggesting a greater improvement in the flow of the prepared granules upon granulation using MADG technique. As shown in Table 7, the results of regression analysis showed that the water amount and the PGS amount had a significant (*p* = 0.0001 for the water amount and *p* = 0.0278 for the PGS amount) negative effect on granules’ angle of repose with respect to the negative sign of their coefficient estimates (−3.21 for the water amount and −1.06 for the PGS amount). However, the water amount was the most influential factor, as shown by its large sum of square (61.89 for the water amount and 6.70 for the PGS amount). The 3D response surface plot shown in Figure 1D represents the inverse effect of the water amount and the PGS amount on the granules angle of repose. As discussed before, the higher water amount and PGS amount resulted in an increase in granule size and density and reduction in the percent fines. This finding appeared to contribute to reducing the angle of repose and improving the flow of the prepared granules [38].

The analysis of variance demonstrated that the linear model is valid with a significant *p*-value (*p* = 0.0003) and insignificant lack of fit value (F = 42.64). The linear model equation that explained the effect of water amount and PGS amount on the angle of repose could expressed as follows:

Angle of repose (degree) = 28.46 − 3.21 × X_1_ − 1.06 × X_2_

### 3.3. Effect of Independent Variables on Tablets’ Characteristics

#### 3.3.1. Tablet Weight Variation

The most important reason for the granulation process is to produce tablets with low weight variation to ensure a uniform drug content, which is related to granules’ flow [39]. Table 8 shows the values of tablets’ average weight and its standard deviation (SD) for all formulations. It was observed that the tablet weight variation was acceptable for all formulations with respect to USP criteria and the SD was less than 2.0, indicating an acceptable flow of the prepared granules. However, the small weight variation of the produced tablets could be owing to the variation in the bulk density of compressed granules [33]. The results of ANOVA given in Table 9 showed that both tested variables had a significant (*p* = 0.0002 for the water amount and *p* = 0.0022 for the PGS amount) negative effect on SD of tablet weight variation that is evident from their negative sign of coefficient estimates (−0.2117 for the water amount and −0.090 for the PGS amount). However, the water amount had the pronounced effect with respect to the magnitudes of its sum of squares (0.2668 for the water amount and 0.0486 for the PGS amount). As shown in Figure 2A, the significant variables have an inverse relationship with SD of tablet weight variation, suggesting that an increase in any of the independent variables individually leads to reduction of SD of tablet weight variation. Moreover, the smallest value of SD was obtained using granules prepared at the combination of a high-water amount and high PGS amount. This might be attributed to the improvement of the granules’ flowability upon increasing the water amount and PGS amount, as previously discussed in Section 3.2.3. It is noteworthy that the angle of repose results showed a high correlation (R^2^ = 0.9538) with the SD of tablet weight variation.

The analysis of variance demonstrated that the quadratic model is valid with a significant *p*-value (*p* = 0.001) and insignificant lack of fit value (F = 131.15). The quadratic model equation that explained the effect of water amount and PGS amount on the SD of tablets weight variation could expressed as follows:

SD of tablet weight variation = 1.53 − 0.2117 × X_1_ − 0.090 × X_2_ + 0.0125 × X_1_X_2_ + 0.065 × X_1_^2^ + 0.040 × X_2_^2^

#### 3.3.2. Mechanical Strength of the Prepared Tablets

As the strength of a tablet might be significantly linked to the drug release behavior when it is administrated in the patient’s body, it is important to measure the tablet mechanical strength using suitable tests, likes breaking force and friability test [39]. The results of tablets’ breaking force for all formulations are shown in Table 8. The breaking force of compressed tablets ranged from 4.18 to 6.97 KP. It was obvious that tablets’ breaking force was positively correlated with the amount of added water between 1% and 2.5% and the breaking force increased from 4.18 to 6.97 KP. However, by increasing the water amount over 2.5%, the breaking force of prepared tablets decreased significantly from 6.97 to 4.76 KP. This might be attributed to the decrease in yield pressure with increasing the water amount over 2.5%. These results indicate that increasing or decreasing the water amount can increase or decrease the tablets’ breaking force. It was reported that free water content has a significant positive effect on MADG tablet tensile strength when the water amount is between 0.0% and 2.5%, while it has a negative impact when the water amount exceeds 2.5% (i.e., tablets’ tensile strength reached a peak and then started to decline when the water amount was approximately doubled) [40]. The results of ANOVA given in Table 9 showed that both variables had a significant (*p* = 0.0128 for the water amount and *p* = 0.0161 for the PGS amount), but mostly equal effect on tablets’ breaking force that is evident from their sum of square values (0.1667 for the water amount and 0.1411 for the PGS amount). As shown in Table 9, the values of coefficient estimates revealed that the change in the water amount and the PGS amount had nearly the same effect on tablet breaking force in the positive direction, which is evident from their positive sign of coefficient estimates (+0.1667 for the water amount and = +0.1533 for the PGS amount). The response surface presented in Figure 2B was curved owing to the presence of the quadratic terms in the proposed model. The curvature of the response surface plot demonstrated that the value of high breaking force lay in middle of upper edge of left wall consisting of a higher amount of PGS and intermediate amount of added water, while lowest value was observed at a lower amount of water and PGS.

Another tablet property related to mechanical strength of the prepared tablets is friability. The results of tablet friability and percent capping for all formulations are given in Table 8. It was obvious that formulations 1, 2, and 3 (low amount of added water) produced tablets with a higher capping tendency. This could be attributed to improper wetting of powder particles and elevation of the percent fines [41]. This finding clearly suggested the inadequacy of these formulations for preparing of tablets with acceptable attributes. The results of the regression analysis given in Table 9 showed that only the water amount had a significant (*p* = 0.0016 for the water amount and *p* = 0.0917 for the the PGS amount) negative impact on percent capping of the prepared tablets with respect to the negative sign of coefficient estimates (−10.0 for the water amount). As shown in Figure 2C, an increase in the water amount led to a decrease in the capping tendency of the prepared tablets. On the other hand, formulations 4–9 produce acceptable tablets in terms of friability (percent loss <1%) and capping with respect to USP limit.

The regression analysis showed that the quadratic models were valid for breaking force and percent capping with a significant *p*-value (*p* = 0.0004 and *p* = 0.0072) and insignificant lack of fit value (F = 264.00 and F = 35.41), respectively. The quadratic equations that explained the effect of the water amount and PGS amount on breaking force and percent capping could expressed as follows:

Breaking force (KP) = 6.8 + 0.1667 × X_1_ + 0.1533 × X_2_ − 0.4250 X_1_X_2_ − 1.82 × X_1_^2^ − 0.0733 × X_2_^2^

Percent Capping (%) = 0.0011 − 10.00 × X_1_ − 2.22 × X_2_ + 3.33 × X_1_X_2_ + 10.00 × X_1_^2^ − 0.0017 × X_2_^2^

#### 3.3.3. In Vitro and In Vivo Disintegration

Table 8 shows the results of tablet disintegration time for all formulations. It was found that the in vitro DT ranged from 33.47 to 92.31 s. The results of ANOVA given in Table 9 demonstrated that the water amount and the PGS amount had a significant (*p* = 0.0002 for the water amount and *p* < 0.0001 for the PGS amount) negative effect on tablet DT with respect to the negative sign of their coefficient estimates (–8.75 for the water amount and –11.13 for the PGS amount). Two-way interaction between the water amount and the PGS amount also had a significant (*p* = 0.0027) positive (coefficient of estimate = +4.24) effect on tablet DT. However, the PGS amount had a pronounced effect on DT according to the magnitude of its sum of squares compared with the water amount and the interaction between the two variables (742.82 for the PGS amount, 459.55 for the water amount, and 71.91 for the variables’ interaction). It was reported that PGS promotes tablet disintegration owing to its ability to produce swelling; PGS quickly absorbs water, resulting in swelling that leads to rapid tablet disintegration [42,43]. The response surface presented in Figure 2D was curved because the model included significant quadratic terms. The curvature of the response surface plot demonstrated that the value of low DT lay in the middle of the lower edge of the left wall, consisting of a higher amount of PGS and an intermediate amount of added water.

As depicted in Figure 3, DT was inversely proportional to the water amount with a rapid and sharp decrease in DT from 92.31 s to 36 s as the water amount increased from 1% to 2.5% at a low amount of PGS (5%). The trend was repeated at a high amount of PGS (15%), with a sharp decrease in DT from 61.84 s to 33.47 s as the water amount increased from 1% to 2.5%. However, further increasing the water amount over 2.5%, the DT was significantly increased from 56.25 s to 65.45 s and from 33.47 s to 51.94 s at a low and high level of PGS, respectively.

The regression analysis displayed that the quadratic model is valid with a significant *p*-value (*p* = 0.0001) and insignificant lack of fit value (*F* = 543.63). The quadratic model equation that explained the effect of water amount and PGS amount on the tablet DT could expressed as follows:

DT (s) = 43.71 − 8.75 × X_1_ − 11.13 × X_2_ + 4.24 × X_1_X_2_ + 22.91 × X_1_^2^ + 1.22 × X_2_^2^

Figure 4 shows the in vivo DT for all runs. It was obvious that DT in the oral cavity of subjects was slower than in the in vitro test. The variation between in vivo and in vitro DT is owing to the existence of the strong agitation carried out by the apparatus over the tablets through the in vitro procedure as well as use a huge amount of disintegration media compared with the in vivo test. This result is in a good agreement with what was reported by Khafagy et al. [33] and Alalaiwe et al. [34]. Figure 5 displays an excellent correlation (R^2^ = 0.9918) between in vivo and in vitro DT. This indicates that the prepared ODT shows rapid disintegration in the oral cavity of human volunteers.

#### 3.3.4. In Vitro Drug Release

Drug dissolution is a key parameter as it governs the rate of the drug release from the tablet, and hence its bioavailability [44]. In vitro drug release profiles for all formulations were characterized and the results are shown in Table 8. It was observed that all ODT prepared by the MADG method showed compliance with the USP limit for the immediate release tablet (i.e., 80% release within 30 min). The results of ANOVA of drug release data after 30 min are given in Table 9. The regression analysis showed that the water amount and the PGS amount had a significant (*p* = 0.0038 for the water amount and *p* = 0.0081 for the PGS amount) positive impact on percent drug release after 30 min with respect to the positive sign of their coefficient estimates (+4.42 for the water amount and +3.40 for the PGS amount). However, the water amount had a predominant effect on percent drug release, according to the magnitude of its sum of squares compared with the PGS amount (117.40 for the water amount and 69.29 for the PGS amount). As shown in Figure 6, both variables had a greater effect on the percent drug release after 30 min in the positive direction. Moreover, the curvature of the 3D response surface plot demonstrated that the value of higher percent drug release lay in the middle of upper edge of the left wall, consisting of a higher amount of PGS and an intermediate amount of added water. These results might be attributed to the improvement of wettability and disintegration of prepared tablets owing to the hydrophilic nature of PGS, as previously described in Section 3.3.3. It is noteworthy that tablets’ disintegration time showed a good correlation (R^2^ = 0.9243) with the percent drug release.

The regression analysis showed that the quadratic model is valid with a significant *p*-value (*p* = 0.0030) and insignificant lack of fit value (*F* = 64.19). The quadratic model equation that explained the effect of water amount and PGS amount on the percent drug release after 30 min could be expressed as follows:

Percent drug release (%) = 97.61 + 4.01 × X_1_ + 2.99 × X_2_ − 1.72 × X_1_X_2_ − 6.04 × X_1_^2^ − 1.11 × X_2_^2^

### 3.4. Optimization of Experimental Design

Generally, the optimization step was applied to define the optimum values of independent variables in order to develop a robust formulation with desired characteristics [31]. Numerical optimization has been applied to develop an optimized formulation by setting constraints on the dependent and independent variables, as shown in Table 10. Optimization showed a confined design space to obtain the highest breaking force, lowest DT, and percent drug release after 30 min more than 90% using a high amount of PGS (14.99%) and medium amount of water (2.13%), with robustness in the results. As shown in Table 11, the experimental values of breaking force (6.71 KP ± 1.33), DT (34.56 s ± 1.21), and percent drug release after 30 min (96.43% ± 2.01%) were in close agreement with the predicted values of breaking force (6.83 KP), DT (36.27 s), and percent drug release after 30 min (98.56%). Additionally, the relative errors between the experimental and predicted values were less than 5% for each dependent response, proving the validity and predictability of the design. Furthermore, the optimized formula showed in vivo DT of 54.32 s ± 4.31, suggesting rapid disintegration in the oral cavity.

## 4. Conclusions

The present investigation can conclude that MADG offers an opportunity to address the poor flowability and tabletability problems of high-dose metformin, which is important for the successful development of ODT, which improves the compliance of diabetic patients. This study demonstrated that the ODTs prepared by MDAG were successfully optimized by applying the quality-by-design approach. The regression analysis showed that the amounts of added water and PGS had a significant (*p* ≤ 0.05) effect on critical attributes of granules and tablets, with a predominant effect of the water amount. However, the amount of PGS had a pronounced influence on tablet disintegration. An acceptable ODT of high-dose metformin was successfully developed based on the MDAG technique. Such an ODT showed high mechanical strength, short oral disintegration, and acceptable release according to USP criteria of ODT. From an industrial perspective, production of tablets using the MADG method reduces the steps of the manufacturing process, as granules obtained after granulation process were directly subjected to compression without drying and milling. This can substantially decrease the cost of manufacturing thanks to the use of a lesser amount of excipients and unit operations.

## Figures and Tables

**Figure 1 pharmaceutics-12-00598-f001:**
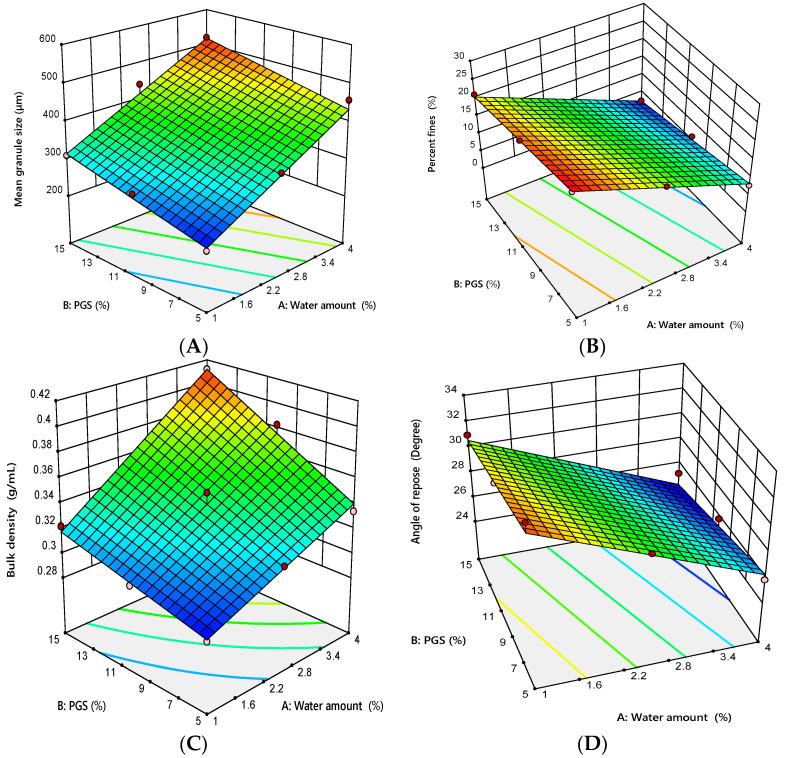
Response surface plot for the influence of the water amount (X_1_) and pregelatinized starch (PGS) amount (X_2_) on granules’ properties, (**A**) mean granule size (**B**), percent fines, (**C**) bulk density, and (**D**) angle of repose.

**Figure 2 pharmaceutics-12-00598-f002:**
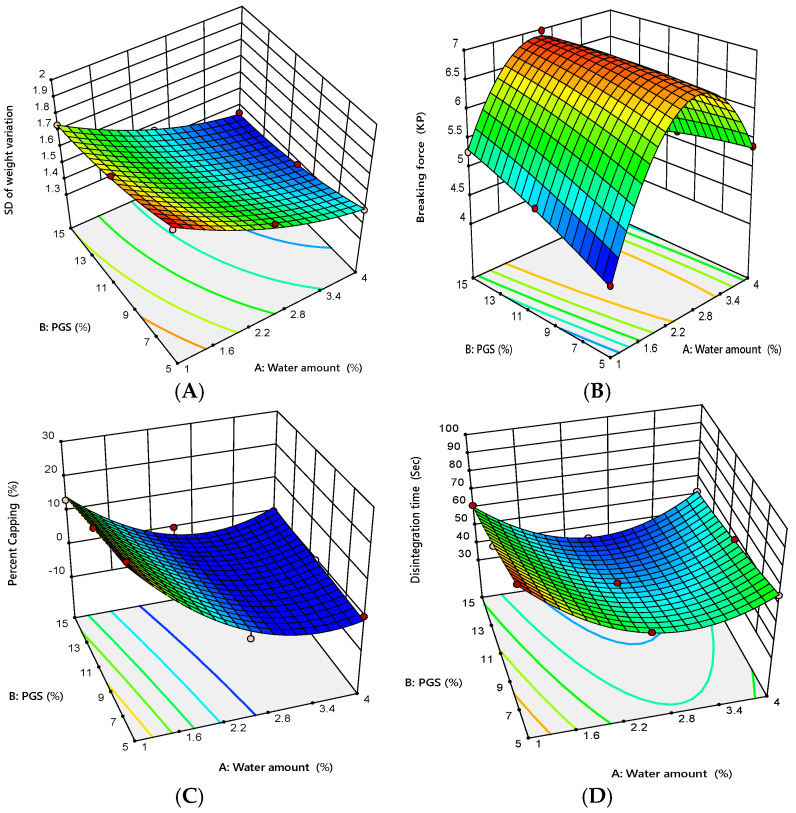
Response surface plot for the influence of water amount (X_1_) and PGS amount (X_2_) on tablets’ properties, (**A**) SD of weight variation, (**B**) breaking force, (**C**) percent capping, and (**D**) in vitro disintegration.

**Figure 3 pharmaceutics-12-00598-f003:**
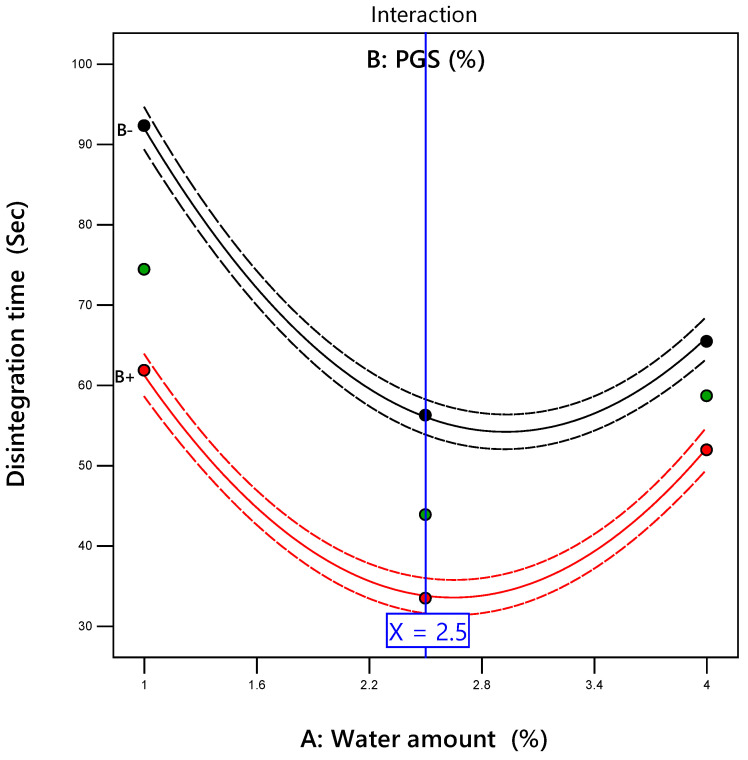
The interaction plot showing the influence of the water amount (X_1_) and PGS amount (X_2_) on in vitro disintegration time of metformin orally disintegrating tablet (ODT).

**Figure 4 pharmaceutics-12-00598-f004:**
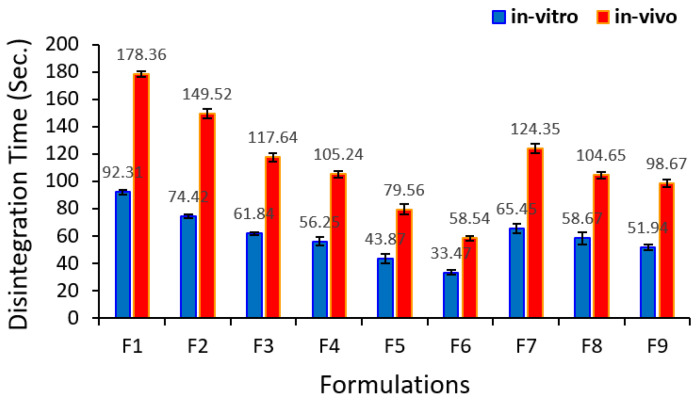
In vitro and in vivo disintegration time (mean ± SD) for all metformin formulations based on the 3^2^ full factorial design.

**Figure 5 pharmaceutics-12-00598-f005:**
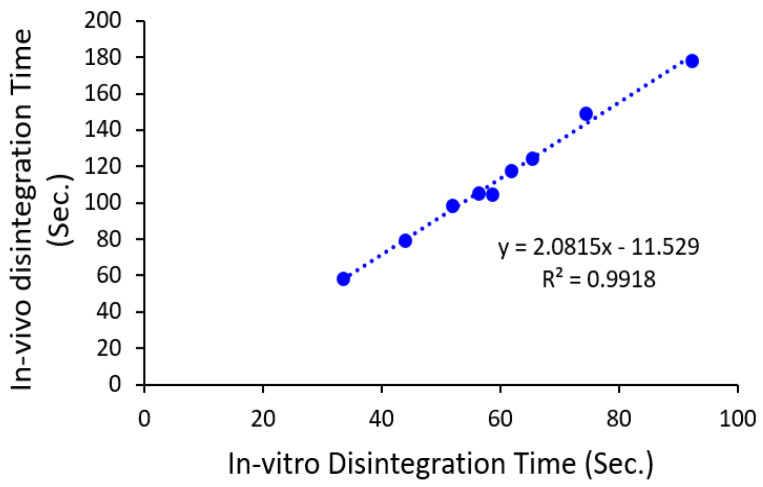
The correlation between in vitro and in vivo disintegration time of all metformin formulations.

**Figure 6 pharmaceutics-12-00598-f006:**
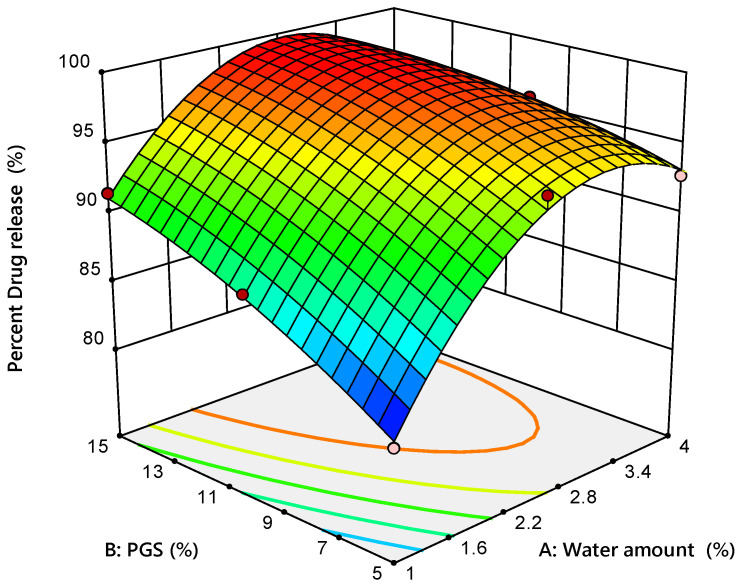
Response surface plot for the influence of water amount (X_1_) and PGS amount (X_2_) on percent release of metformin after 30 min.

**Table 1 pharmaceutics-12-00598-t001:** Quality target product profile (QTPP) and critical quality attributes (CQAs) of metformin fast orally disintegrating tablets. USP, United States Pharmacopeia.

QTPP Element	Target	CQAs	Target
Dosage form	Orally disintegrating tablets	Breaking force	Hard enough
Appearance	Uncoated tablets	Friability	<1%
Strength Route of administration Proposed indications Dosage frequency	500 mg Oral Type-2 diabetes Twice daily	Disintegration time Drug release - -	<60 s Not less than 80% in 30 min (USP) -

**Table 2 pharmaceutics-12-00598-t002:** The selected levels of independent variables used in design of experiment (DoE). PGS, pregelatinized starch.

Coded Levels	Water Amount (%)	PGS Amount (%)
−1	1	5
0	2.5	10
1	4	15

−1: factor at low level; 0: factor at medium level; 1: factor at high level.

**Table 3 pharmaceutics-12-00598-t003:** A full matrix of (3^2^) full factorial design for independent variables.

Experiment Code	Water Amount (%)	PGS Amount (%)
1	1	5
2	1	10
3	1	15
4	2.5	5
5	2.5	10
6	2.5	15
7	4	5
8	4	10
9	4	15

**Table 4 pharmaceutics-12-00598-t004:** The quantitative composition of metformin hydrochloride formulation.

Ingredients	% *w*/*w*
Metformin HCL	80
Pre-gelatinized starch (PGS)	5, 10, 15
d-mannitol	Up to 100
Colloidal silicon dioxide	1.5
Sodium stearyl fumarate	1

**Table 5 pharmaceutics-12-00598-t005:** Model summary statistics and model selection criteria for dependent responses.

Response	Model	*p*-Value	R^2^	Adjusted R^2^	Predicted R^2^	Significance
D_50_	**Linear**	**<0.0001**	**0.9603**	**0.9470**	**0.9184**	**Suggested**
	2FI	0.6181	0.9624	0.9398	0.8732	
	Quadratic	0.4965	0.9764	0.9371	0.7371	
	Cubic	0.4676	0.9948	0.9588	0.0606	Aliased
Percent fines	**Linear**	**<0.0001**	**0.9948**	**0.9931**	**0.9885**	**Suggested**
	2FI	0.9051	0.9948	0.9918	0.9796	
	Quadratic	0.5826	0.9964	0.9904	0.9562	
	Cubic	0.0153	1.0000	1.0000	0.9998	Aliased
Bulk density	Linear	0.0003	0.9358	0.9144	0.8235	
	**2FI**	**0.0395**	**0.9746**	**0.9594**	**0.9128**	**Suggested**
	Quadratic	0.6087	0.9818	0.9514	0.7985	
	Cubic	0.4866	0.9957	0.9655	0.2141	Aliased
Angle of repose	**Linear**	**0.0003**	**0.9343**	**0.9124**	**0.8583**	**Suggested**
	2FI	0.3912	0.9441	0.9106	0.7845	
	Quadratic	0.1953	0.9812	0.9498	0.7886	
	Cubic	0.4481	0.9962	0.9698	0.3114	Aliased
SD of weight variation	Linear	<0.0001	0.9584	0.9445	0.9161	
	2FI	0.6466	0.9603	0.9364	0.8298	
	**Quadratic**	**0.0388**	**0.9954**	**0.9879**	**0.9534**	**Suggested**
	Cubic	0.6437	0.9981	0.9849	0.6561	Aliased
Breaking force	Linear	0.8849	0.0399	−0.2801	−1.1228	
	2FI	0.4950	0.1337	−0.3861	−2.4253	
	**Quadratic**	**0.0001**	**0.9977**	**0.9940**	**0.9748**	**Suggested**
	Cubic	0.4791	0.9995	0.9958	0.9052	Aliased
Percent capping	Linear	0.0248	0.7083	0.6111	0.3338	
	2FI	0.3558	0.7583	0.6133	0.0809	
	**Quadratic**	**0.0181**	**0.9833**	**0.9556**	**0.7968**	**Suggested**
	Cubic	0.0004	1.0000	1.0000	1.0000	Aliased
Disintegration time	Linear	0.1133	0.5162	0.3549	−0.0463	
	2FI	0.5847	0.5470	0.2753	−0.7871	
	**Quadratic**	**0.0001**	**0.9989**	**0.9971**	**0.9867**	**Suggested**
	Cubic	0.1456	1.0000	0.9998	0.9957	Aliased
Drug release—30 min	Linear	0.0519	0.6270	0.5026	0.1387	
	2FI	0.4231	0.6762	0.4820	−0.4706	
	**Quadratic**	**0.0048**	**0.9907**	**0.9753**	**0.8901**	**Suggested**
	Cubic	0.2629	0.9994	0.9949	0.8833	Aliased

**Table 6 pharmaceutics-12-00598-t006:** Physical characteristics of prepared granules of 3^2^ full factorial design (mean ± SD).

Formula	Mean Granule Size (µm ± SD)	Percent Fines (% ± SD)	Bulk Density (gcm^−3^ ± SD)	Angle of Repose (Degree ± SD)
1	221.15 ± 0.33	26.42 ± 0.091	0.288 ± 0.028	33.45 ± 0.432
2	281.36 ± 0.51	23.56 ± 0.073	0.301 ± 0.008	31.64 ± 0.551
3	310.18 ± 0.73	21.14 ± 0.053	0.322 ± 0.037	31.01 ± 0.126
4	335.29 ± 0.48	17.67 ± 0.043	0.316 ± 0.019	29.61 ± 0.621
5	345.36 ± 0.64	14.37 ± 0.032	0.349 ± 0.007	27.11 ± 0.112
6	445.31 ± 0.39	10.24 ± 0.015	0.354 ± 0.054	26.45 ± 0.323
7	456.82 ± 0.56	7.83 ± 0.029	0.334 ± 0.014	25.87 ± 0.245
8	446.53 ± 0.93	6.21 ± 0.054	0.383 ± 0.062	25.83 ± 0.621
9	520.39 ± 0.58	2.36 ± 0.027	0.412 ± 0.017	25.13 ± 0.173

**Table 7 pharmaceutics-12-00598-t007:** Analysis of variance (ANOVA) summary and regression analysis of granules dependent responses. CI, confidence interval.

Variables	Coefficient Estimate	Sum of Squares	Standard Error	F-Value	*p*-Value	95% CI Low	95% CI High
**Mean Granule Size** **(Linear Model)**							
Model	**-**	**-**	**-**	**72.53**	**<0.0001**	**-**	**-**
Intercept	373.16	**-**	7.50	**-**	**-**	354.81	391.50
X_1_	101.84	62,230.35	9.18	123.02	**<0.0001**	79.37	124.31
X_2_	43.11	11,150.83	9.18	22.04	**0.0033**	20.64	65.58
**Percent Fines** **(Linear Model)**							
Model	**-**	**-**	**-**	**577.35**	**<0.0001**	**-**	**-**
Intercept	14.42	**-**	0.2309	**-**	**-**	13.86	14.99
X_1_	−9.12	499.05	0.2828	1039.91	**<0.0001**	−9.81	−8.43
X_2_	−3.03	55.09	0.2828	114.79	**<0.0001**	−3.72	−2.34
**Bulk Density** **(2FI Model)**							
Model	**-**	**-**	**-**	**64.06**	**0.0002**	**-**	**-**
Intercept	0.3399	**-**	0.0027	**-**	**-**	0.3331	0.3467
X_1_	0.0363	0.0079	0.0032	125.24	**<0.0001**	0.0280	0.0447
X_2_	0.0250	0.0038	0.0032	59.29	**0.0006**	0.0167	0.0333
X_1_X_2_	0.0110	0.0005	0.0040	7.65	**0.0395**	0.0008	0.0212
**Angle of Repose** **(Linear Model)**							
Model	**-**	**-**	**-**	**42.64**	**0.0003**	**-**	**-**
Intercept	28.46	**-**	0.2989	**-**	**-**	27.72	29.19
X_1_	−3.21	61.89	0.3661	76.95	**0.0001**	−4.11	−2.32
X_2_	−1.06	6.70	0.3661	8.33	**0.0278**	−1.95	−0.1608

X_1_ and X_2_ represent the amount of added water and the PGS amount, respectively; X_1_X_2_ is the effect of interaction.

**Table 8 pharmaceutics-12-00598-t008:** Quality attributes of prepared metformin hydrochloride orally disintegrating tablets (mean ± SD).

Formula	Weight (mg ± SD)	Thickness (mm ± SD)	Breaking Force (KP ± SD)	Friability (% ± SD)	Percent Capping (%)	Disintegration Time (Sec ± SD)	% Release at 30 min (% ± SD)
1	622.61 ± 1.93	4.33 ± 0.023	4.18 ± 0.43	NA	26.66 ± 5.77	92.31 ± 2.31	81.24 ± 3.11
2	624.64 ± 1.82	4.31 ± 0.003	4.84 ± 0.31	NA	20 ± 0.00	74.42 ± 1.52	87.74 ± 1.68
3	626.91 ± 1.74	4.35 ± 0.008	5.27 ± 0.64	NA	13.33 ± 5.77	61.84 ± 2.23	91.48 ± 1.87
4	623.91 ± 1.68	4.32 ± 0.03	6.53 ± 0.68	0.57 ± 0.04	No capping	56.25 ± 1.41	94.47 ± 2.46
5	621.82 ± 1.47	4.31 ± 0.005	6.76 ± 0.84	0.51 ± 0.06	No capping	43.87 ± 2.32	97.35 ± 2.91
6	624.45 ± 1.50	4.37 ± 0.006	6.97 ± 0.86	0.39 ± 0.03	No capping	33.47 ± 1.42	98.79 ± 2.73
7	623.37 ± 1.49	4.33 ± 0.008	5.37 ± 0.48	0.70 ± 0.03	No capping	65.45 ± 1.62	92.75 ± 2.67
8	622.51 ± 1.38	4.32 ± 0.004	5.16 ± 0.62	0.64 ± 0.07	No capping	58.67 ± 2.16	95.67 ± 3.11
9	626.47 ± 1.35	4.32 ± 0.04	4.76 ± 0.34	0.51 ± 0.05	No capping	51.94 ± 1.25	96.12 ± 2.49

**Table 9 pharmaceutics-12-00598-t009:** Analysis of variance (ANOVA) summary and regression analysis of tablets’ dependent responses.

Variables	Coefficient Estimate	Sum of Squares	Standard Error	F-Value	*p*-Value	95% CI Low	95% CI Low
**SD of Weight Variation** **(Quadratic Model)**							
Model	-	**-**	-	**131.15**	**0.001**	-	-
Intercept	1.56	**-**	0.0130	**-**	**-**	1.53	1.59
X_1_	−0.1617	0.1568	0.0159	103.14	**<0.0001**	−0.2006	−0.1227
X_2_	−0.0717	0.0308	0.0159	20.27	**0.0041**	−0.1106	−0.0327
X_1_X_2_	13.33	25.00	1.27	3.86	0.1443	−1.55	6.55
X_1_^2^	−1.67	355.56	1.80	54.86	**0.0051**	7.60	19.06
X_2_^2^	−13.33	5.56	1.80	0.8571	0.4228	−7.40	4.06
**Breaking Force** **(Quadratic Model)**							
Model	-	**-**	-	**264.00**	**0.0004**	-	-
Intercept	6.27	**-**	0.0549	**-**	**-**	6.14	6.40
X_1_	−0.4867	1.42	0.0673	52.34	**0.0004**	−0.6513	−0.3221
X_2_	−0.4183	1.05	0.0673	38.67	**0.0008**	−0.5829	−0.2537
X_1_X_2_	13.33	25.00	1.27	3.86	0.1443	−1.55	6.55
X_1_^2^	−1.67	355.56	1.80	54.86	**0.0051**	7.60	19.06
X_2_^2^	−13.33	5.56	1.80	0.8571	0.4228	−7.40	4.06
**Percent Capping** **(Quadratic Model)**							
Model	-	**-**	-	**35.41**	**0.0072**	-	-
Intercept	−13.33	**-**	1.90	**-**	**-**	−4.93	7.15
X_1_	−1.67	1066.67	1.04	164.57	**0.0010**	−16.64	−10.03
X_2_	2.50	16.67	1.04	2.57	0.2071	−4.97	1.64
X_1_X_2_	13.33	25.00	1.27	3.86	0.1443	−1.55	6.55
X_1_^2^	−1.67	355.56	1.80	54.86	**0.0051**	7.60	19.06
X_2_^2^	−13.33	5.56	1.80	0.8571	0.4228	−7.40	4.06
**Disintegration Time** **(Quadratic Model)**							
Model	-	**-**	-	**543.63**	**0.0001**	-	-
Intercept	9.94	**-**	0.2140	**-**	**-**	9.25	10.62
X_1_	4.49	120.87	0.1172	1466.81	**<0.0001**	4.12	4.86
X_2_	1.15	7.96	0.1172	96.57	**0.0022**	0.7787	1.52
X_1_X_2_	0.6650	1.77	0.1435	21.47	**0.0189**	0.2082	1.12
X_1_^2^	2.01	8.09	0.2030	98.22	**0.0022**	1.37	2.66
X_2_^2^	0.1517	0.0460	0.2030	0.5583	0.5092	−0.4943	0.7976
**Percent Release at 30 min** **(Quadratic Model)**							
Model	-	**-**	-	**64.19**	**0.003**	-	-
Intercept	83.96	**-**	0.5714	**-**	**-**	82.56	85.36
X_1_	−5.86	206.27	0.6998	70.21	**0.0002**	−7.58	−4.15
X_2_	−2.81	47.38	0.6998	16.12	**0.0070**	−4.52	−1.10
X_1_X_2_	13.33	25.00	1.27	3.86	0.1443	−1.55	6.55
X_1_^2^	−1.67	355.56	1.80	54.86	**0.0051**	7.60	19.06
X_2_^2^	−13.33	5.56	1.80	0.8571	0.4228	−7.40	4.06

X_1_ and X_2_ represent the amount of water and the wet massing time, X_1_X_2_ is the effect of interaction, and X_1_^2^ and X_2_^2^ are the sum of effects.

**Table 10 pharmaceutics-12-00598-t010:** The constraints adopted for the optimization of tested variables and estimation of overall desirability.

Variables	Target	Range	Weight	Importance Co-Efficient
**Input**				
Water amount	In range	1–4%	1	-
PGS amount	In range	5–15%	1	-
**Output**				
Breaking force Percent capping	Maximize No capping	4.18–6.97 KP 0–26.66%	1	++++ ++++
Disintegration time	Minimize	33.47–92.31 s	1	++++
Percent release at 30 min	In range	81.24–98.79%	1	++++

**Table 11 pharmaceutics-12-00598-t011:** Predicted and observed values for all dependent responses of optimized formulation with their relative errors, overall desirability = 0.968 (close to 1).

Responses	Predicted Values	Observed Values (Mean ± SD)	Relative Error (%)
Breaking force (KP)	6.83	6.71 ± 1.33	1.75
Percent capping (%)	No capping	No capping	Zero%
Disintegration time (Sec)	36.27	34.56 ± 1.21	4.71
Percent release at 30 min	98.56	96.43 ± 2.01	2.16

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
