# Peer review of "Design, Optimization, and Correlation of In Vitro/In Vivo Disintegration of Novel Fast Orally Disintegrating Tablet of High Dose Metformin Hydrochloride Using Moisture Activated Dry Granulation Process and Quality by Design Approach"

_pharmaceutics, 2020, doi:10.3390/pharmaceutics12070598_

Round 1
Reviewer 1 Report
Summary: The authors have presented an optimization of ODT using MADG for a challenging drug substance and the manuscript is well written. Although the authors have presented data regarding feasibility of using MADG for ODT, in my opinion, the authors have mainly done a fitting exercise of their data. The fundamental understanding of why the observed effects are seen is lacking in this manuscript. The result that water added and level of PGS has a significant effect on the tablet and granule properties is not new- it is a well known result in the area of wet granulation and I believe MADG is not very different.
- The MADG process seems very similar to a high shear wet granulation process. Can the authors add a paragraph in the introduction section explaining key differences between the two processes, other than using a smaller value of water added compared to the wet granulation method?
- In the materials and methods section, please specify the tooling dimensions used for pressing the tablets.
- For an orally disintegrating tablet, why did the authors not consider using a disintegrate like croscarmellose sodium in the formulation? Similarly, why did the authors not use a mildly disintegrating excipient with superior mechanical properties such as Avicel instead of using Mannitol? What is the specific purpose of using mannitol?
- The increase in breaking force for tablets with increase in water added followed by decrease in breaking force with further water added is not clear to me. Considering that all tablets were made using the same compression force, the authors should plot a tensile strength vs solid fraction curve for each formulation in consideration to better understand this effect.
- How is the % capping measured?
- In section 3.3.3., the authors mention that the difference in the in-vitro and in-vivo disintegration time was due to mechanical effect in the in-vitro procedure and huge amount of disintegration media in the in-vitro procedure. Can the authors elaborate on what mechanical effect they are referring to?
- Most of the observed disintegration times in the oral cavity of the volunteers are longer than a minute. What is the optimum target disintegration time that is desired? Have the authors considered optimization of tablet tensile strength or solid fraction to further reduce this disintegration time?
- What is the level of residual moisture in the tablet? Is this moisture expected to cause any undesired effects- microbial growth, stability issues in the tablet?
Author Response
Response to the academic editor:
Thank you very much for your valuable comment. We greatly appreciate your great effort in reviewing the present research. Please note that the added or changed part had been highlighted in the text.
Reviewer’s comment:
"It should be taken into consideration that the title mentions "Quality by design", but no such concept and paradigm was found explicit or implicit within the manuscript!"
Authors’ response:
Quality by Design approach had been explained and added to the text. Section 1. Lines 106 – 110.
Response to reviewer 1:
Thank you very much for your valuable comments. We greatly appreciate your great effort in reviewing the present research. Please note that the added or changed part had been highlighted in the text.
- Reviewer’s comment:
The MADG process seems very similar to a high shear wet granulation process. Can the authors add a paragraph in the introduction section explaining key differences between the two processes, other than using a smaller value of water added compared to the wet granulation method?
Authors’ response:
Key differences between high-shear granulation and moist activated dry granulation had been explained and added to the introduction part. Section 1. Lines 88 – 98.
- Reviewer’s comment:
In the materials and methods section, please specify the tooling dimensions used for pressing the tablets.
Authors’ response:
Tooling dimensions used for tablets compression had been added to the granules and tablets preparation part. Section 2.4. Lines 162 – 163.
- Reviewer’s comment:
For an orally disintegrating tablet, why did the authors not consider using a disintegrate like croscarmellose sodium in the formulation? Similarly, why did the authors not use a mildly disintegrating excipient with superior mechanical properties such as Avicel instead of using Mannitol? What is the specific purpose of using mannitol?
Authors’ response:
Pregelatinized starch was chosen in the present study because it shows dual functionality as binder/ disintegrant. It could exhibit binding properties in the granulation applications, then it would become strong disintegrants when exposed to water due to its ability to induce swelling. In addition, Pregelatinized starch had a lower propensity for moisture uptake than sodium starch glycolate and croscarmellose sodium and drew less moisture into tablets. Thus it is a suitable binder/ disintegrant for moisture sensitive drugs (i.e. metformin) in granulation applications.
Ali R. Rajabi-Saihboomi et al. Chapter 13: Excipient selection in oral solid dosage formulations containing moisture-sensitive drugs. In: Ajit S. Naraj and Sai HS Boddu editors. Excipient applications in formulation design and drug delivery. Springer 2015 p: 403.
Mannitol is a preferred excipient for developing ODT of the moisture-sensitive drugs like metformin due to its non-hygroscopic nature, compactibility, sweet taste, cool feeling that it leaves in the mouth. Besides, it reduces disintegration time in the oral cavity due to its higher solubility compared to other water-soluble excipients used in the preparation of ODT.
Sushma V. Lute, Ranjit M. Dhenge and Agba D. Salman. Twin Screw granulation: Effects of properties of primary powders. Pharmaceutics. 2018, 10, 68.
Solaiman A, Suliman AS, Shinde S, Naz S, Elkordy AA. Application of general multilevel factorial design with formulation of fast disintegrating tablets containing croscarmellose sodium and Disintequick MCC-25. Int. J. Pharm. 2016, 501(1-2), 87-95.
Rationale for excipients selection had been explained and added to the text. Section 2.3. Lines 134 - 150.
- Reviewer’s comment:
The increase in breaking force for tablets with increase in water added followed by decrease in breaking force with further water added is not clear to me. Considering that all tablets were made using the same compression force, the authors should plot a tensile strength vs solid fraction curve for each formulation in consideration to better understand this effect.
Authors’ response:
It was reported that water activity has a positive impact on MADG tablet tensile strength between 0.0% and 2.5% added water. By increasing the amount of added water over 2.5%, the tensile strength of tablets prepared by MADG granules decreased significantly between 2.5% and 5.0% added water. MADG tablet tensile strength was negatively correlated with water activity between 2.5% and 5.0% (i.e. tensile strength reached a peak and started to decrease when moisture content was approximately doubled). These results suggest increasing or decreasing the amount of moisture can increase or decrease the tensile strength, with the exact change dependent on the moisture content of the powders. Section 3.3.2. Lines 351 – 355.
Takasaki et al. The effect of water activity on granule characteristics and tablet properties produced by moisture activated dry granulation (MADG). Powder Technol. 2016;294:113-8.
- Reviewer’s comment:
How is the % capping measured?
Authors’ response:
Upon dusting the tablets following friability testing, the number of tablets that showed capping was determined. Capping of the tablets was reported as the percent of tablets capped out of the total tested tablets (10 tablets). Section 2.6.3. Lines 197 – 199.
- Reviewer’s comment:
In section 3.3.3., the authors mention that the difference in the in-vitro and in-vivo disintegration time was due to mechanical effect in the in-vitro procedure and huge amount of disintegration media in the in-vitro procedure. Can the authors elaborate on what mechanical effect they are referring to?
Authors’ response:
Mechanical effect is due to the strong agitation carried out by the apparatus over the tablets through the in-Vitro procedure. This clarification had been added to the text. Section 3.3.3. Lines 415 - 416.
- Reviewer’s comment:
Most of the observed disintegration times in the oral cavity of the volunteers are longer than a minute. What is the optimum target disintegration time that is desired? Have the authors considered optimization of tablet tensile strength or solid fraction to further reduce this disintegration time?
Authors’ response:
According to the FDA guidance for industry (2008), ODT is considered solid oral preparations that disintegrate rapidly in the oral cavity, with an in-vitro disintegration time of approximately 30 seconds or less, when based on the United States Pharmacopeia disintegration test method or alternative. However; the Ph. Eur. the limit is within 3 min.
Vanbillemont et al. New advances in the characterization of lyophilized orally disintegrating tablets. Int. j. Pharm. 2020, 579, 119153.
To maintain ODT for high dose drugs we defined a target value with less than 60 s as a mean value. Optimization showed a confined design space to obtain the highest mechanical strength, lowest DT in patient’s mouth of less than 60 s with robustness in results. The optimized formula compressed at 13 KN showed in-vivo DT of 54.32 Sec ± 4.31 suggesting rapid disintegration in the oral cavity. Section 3.4. Lines 457 – 458.
- Reviewer’s comment:
What is the level of residual moisture in the tablet? Is this moisture expected to cause any undesired effects- microbial growth, stability issues in the tablet?
Authors’ response:
The level of residual moisture for all runs (1 - 9) was ranging from 1.36 to 3.64 %. The stability of the prepared tablets as well as in-vivo study for optimized formula will be further investigated. However, according to levels of moisture content and formulation composition, the stability of prepared tablets is expected. PGS had a lower propensity for moisture uptake and drew less moisture into tablets. Thus, PGS had a significantly higher positive impact on product stability. Besides, mannitol is a preferred excipient for developing ODT of moisture-sensitive drugs like metformin due to its non-hygroscopic nature.

Reviewer 2 Report
Dear,
thank you for submitting this paper entitled 'Design and optimization of a novel fast orally disintegrating tablet of high-dose metformin hydrochloride using moisture-activated dry granulation process and quality by design approach' which shows an interesting approach to create a wet granulation process by easily removing the added moisture after the process using the MADG technique. The new formulations were critically characterized and I have only some minor suggestions/remarks:
*introduction: '3gm' should be '3 g'
*you are mentioning an absorbent powder to remove the moisture but I believe it should be an adsorbent powder as this is a adsorption process and not an absorption process
*materials and methods: page 6: Granulation experiments was were
*Related to the dissolution experiments: I don not believe that these dissolution tests are very biorelevant (working with 800/900 mL of dissolution media, etc.) but are rather taken from standard quality control experiments (US Pharmacopeia) - am I right? Please mention that biorelevance for these dissolution experiments lack but that, based on the comparative study that you want to make, that this is not a major issue with respect to the purpose.
*I'm really suprised by the clinical study that was performed and the IVIVC that was observed; which is a really good thing and interesting! you should be proud of that and maybe reflect the IVIVC also in the title of this manuscript to make it more attractive for people to read this! It's not that common that technologists set up a clinical study for their formulations, but I'm really positive about this!
Author Response
Response to reviewer 2:
Thank you very much for your valuable comments. We greatly appreciate your great effort in reviewing the present research. Please note that the added or changed text has been highlighted with yellow color.
- Reviewer’s comment:
Introduction: '3gm' should be '3 g
Authors’ response:
The required change had been done in the text. Section1. Line 59.
- Reviewer’s comment:
You are mentioning an absorbent powder to remove the moisture, but I believe it should be an adsorbent powder as this is an adsorption process and not an absorption process.
Authors’ response:
According to the previously reported work, the colloidal silicon dioxide was used as absorbent powder as it absorbs excess water from the wet mass during the absorption stage.
Takasaki et al. The effect of water activity on granule characteristics and tablet properties produced by moisture activated dry granulation (MADG). Powder Technol. 2016;294:113-8.
- Reviewer’s comment:
Materials and methods: page 6: Granulation experiments was were
Authors’ response:
The required change had been done in the text. Section 2.4. Line 153.
- Reviewer’s comment:
Related to the dissolution experiments: I do not believe that these dissolution tests are very biorelevant (working with 800/900 mL of dissolution media, etc.) but are rather taken from standard quality control experiments (US Pharmacopeia) - am I right? Please mention that biorelevance for these dissolution experiments lack but that, based on the comparative study that you want to make, that this is not a major issue with respect to the purpose.
Authors’ response:
Really, the present dissolution procedure was carried out according to the USP and not biorelevant to the oral cavity. The required statement had been added to the text. Section 2.6.6. Lines 220 – 221.
- Reviewer’s comment
I am really surprised by the clinical study that was performed and the IVIVC that was observed; which is a really good thing and interesting! you should be proud of that and maybe reflect the IVIVC also in the title of this manuscript to make it more attractive for people to read this! It's not that common that technologists set up a clinical study for their formulations, but I'm really positive about this!
Authors’ response:
Thank you very much for your kind and valuable comment. IVIVC had been added to the manuscript title.
Response to reviewer 3:
- Reviewer’s comment
Please improve the structure and composing of the article.
Authors’ response:
The structure and composing of the manuscript had been improved.
- Reviewer’s comment
Please briefly introduce what is “A 2-factor 3-levels full factorial design (32)” , and why it was chosen?
Authors’ response:
The rationale for the selection of full factorial design in the present research had been demonstrated in the text. Section 2.2. Lines 118 – 122.
- Reviewer’s comment
Please add the amount of pregelatinized starch and water amount to the formula in Table 5.
Authors’ response:
The amount of pregelatinized starch and water amount had been added to Table 5.

Round 2
Reviewer 1 Report
no additional comments
Author Response
Response to reviewer 1:
Thank you very much for your valuable comments. We greatly appreciate your great effort in reviewing the present research. Please note that the added or changed part had been highlighted in the text.
Reviewer’s comment:
It is our opinion that a few issues need to be resolved before the acceptance of the manuscript. The authors included a few sentences with regard to the Quality by design (QbD) in order to address our first comment on the "abusive" use of the QbD word. Apparently they still missed the point about QbD and it is the first time I read a manuscript about the use of QbD where no single mention to the QTTP (Quality Target Product Profile), to the CQA's (Critical Quality attributes), CMA's (Critical Material Attributes), CPP's (critical process parameters) as well as to the strategy and rationale used to identify them. There is plenty of information available on the subject as well as ICH Q guidelines (mandatory in several drug regulatory frameworks) that cover these issues, which were not referenced. Without proper support (by including additional information) the use of the "and quality by design approach" is abusive and not justified.
Authors’ response:
Additional information regarding the QbD had been added to the text. Section 1, lines 110 – 123.
Reviewer’s comment:
It is apparent, from the manuscript, that no replicate batches of granulated blends were prepared and compressed. Therefore, only 9 tablet batches were prepared, each one corresponding to each of the two factors, 3-level combinations). As such, there is no possibility to directly identify the contribution of the experimental (weight, mix, granulation, compression) variability to the overall measured variability and to estimate the model's uncertainty. In addition, it precludes the proper validation of the model and consequently part of the data analysis comments and conclusions.
Authors’ response:
Confirmatory runs within the design space had been done but the data not shown in the manuscript. The run at the center point was repeated five times at several days, the average values of these experiments show good reproducibility of the process. Section 2.2, line 144 – 146.
Reviewer’s comment:
"s" is the symbol representing the unit "second", not "Sec". Second may be abbreviated to "sec".
Authors’ response:
“Sec” had been replaced with the symbol “s” for all values of disintegration time.
Reviewer’s comment:
In addition: 045: replace "Metformin is classified as class-III drug" by "Metformin is a BCS class-III drug" (or equivalent sentence mentioning "BCS" that stands for Biopharmaceutical Classification System);
Authors’ response:
The required change had been done. Section 1, line 57.
Reviewer’s comment:
173: Clarify the distinction between the "2.6.2. Tablets Breaking Force" and "2.6.3. Tablets Friability and Percent Capping" in particular the friability as according to the manuscript they both reflect the strength of tablets.
Authors’ response:
Breaking force and friability are important to measure tablet mechanical strength. According to the United States Pharmacopeia (chapter 1216 & 1217), the percentage weight loss after tumbling is referred to as the friability of the tablets. Another measure of the mechanical integrity of tablets is their breaking force, which is the force required to cause them to fail (i.e., break) in a specific plane.
Reviewer’s comment:
211: Missing symbols "=" and "+" in the equation 233: in Table 5 the values in the "Water amount (%)" and "PGS amount (%)" columns make no sense
Authors’ response:
The required change had been done in section 3.1, line 244. In addition, Table 5 had been modified and changed to Table 6.
Reviewer’s comment:
263: The friability test is an "official" test in several pharmacopeias and the authors must indicate which pharmacopeia was followed to perform the test (or if any change was made to the official procedure of such pharmacopeia).
Authors’ response:
The friability test had been carried out according to USP 42-NF37 (2019). Section 2.6.3, line 208.
Reviewer’s comment:
302: "was acceptable for all formulations with respect to USP criteria…" Indicate the USP test and/or chapter number where the reader may find the test description and criteria.
Authors’ response:
The test had been carried out according to USP 42-NF37 (2019). Section 2.6.1, line 198.
Reviewer’s comment:
454: Replace "Percent release-30min" by "Percent release at 30 min"
Authors’ response:
The required change had been done. Tables 8, 9, 10, and 11.
Reviewer’s comment:
460: "PGS had a significant (P ≤ 0.5)" Is the sentence correct or should it be (P ≤ 0.05)"? We think that after properly addressing the above issues the manuscript is acceptable for publication.
Authors’ response:
The required change had been done. Section 4, line 481.
